# A Methodology to Model Integrated Smart City System from the Information Perspective

**Jules Muvuna ***, **Tuleen Boutaleb**, **Keith J Baker and Slobodan B Mickovski**

School of Computing Engineering and Built Environment, Glasgow Caledonian University, Cowcaddens Rd, Glasgow G4 0BA, UK; T.Boutaleb@gcu.ac.uk (T.B.); Keith.Baker@gcu.ac.uk (K.J.B.); Slobodan.Mickovski@gcu.ac.uk (S.B.M.)

**\*** Correspondence: Jules.Muvuna@gcu.ac.uk

**Abstract:** Rapid urban population growth challenges cities and sustainable urban development. Despite the effort deployed with conventional urban design, the current solutions are unable to significantly respond to existing challenges. The concept of smart city (SC) has been gaining popularity and cities have developed strong interest for transformation into SCs. However, given that a city is both a complex system of systems and a dynamic complex environment, achieving a state of SC can be challenging. Urban transformation into smart city has been straggling and has shown contrasting and disconnected views. Many studies have covered the design and modelling of an SC, but their focus has been mostly thematic and lack an integrated view of a smart city system. This study presents a methodology helped by a Model-Based Systems Engineering (MBSE) approach and Systems Modelling Language (SysML) to develop a model of an integrated SC system. The model brings all subsystems to operate together in one system and focuses on the information perspective of a city system. Three scenarios are presented to illustrate how an integrated information platform can be a gateway and easy access to information in an SC system, as well as a starting point towards modelling an integrated SC system.

**Keywords:** Smart city; SysML; MBSE; integrated information platform

---

## 1. Introduction

Although globalisation has connected cities and improved collaboration among big cities, it also created competition among them. It is in the interest of every city to increase productivity and innovation with the aim of making itself desirable and attractive for living. While the observed urban population can come with benefits, such as increased labour market, it also comes with challenges, such as waste management, traffic congestion, pollution, surcharge of energy demand, etc. [1]. All this threatens sustainable urban development and creates a demand for efficient and innovative measures to deal with these challenges.

Technology has a major role in cities transformation and is a major SC enabler. The SC concept is by far built on information, with the main features being Information Technology (IT) and comprehensive application of information resources [2]. It can be deducted from [3] that an SC uses intelligent information systems to achieve a sustainable urban development—a definition that places information as a principle factor of the SC concept.

This concept has gained popularity and transformation of existing cities into SCs has been attracting significant attention from various stakeholders [1] A city is a complex system of systems [4] and the rate at which cities are becoming complex was found to be faster than the rate at which theories were being developed to understand them [5]. To cope with cities' complexity, their systems have been divided into small individual systems that address issues thematically [6]. Although divisions of a city

system into individual systems might bring solutions, it will definitely also make the city system even more complex. An SC system should achieve integration and efficiency [7]. To redesign cities into smarter cities means to transform them into more sustainable, integrated, and collaborative at all levels. A city can be considered smarter than another based on how efficient and integrated its subsystems are, as well as the degree of seamless flow of information among subsystems and the general public.

Many attempts have been made to redesign, model, and transform cities—the most complex man-made systems of systems [8]—into smarter systems. However, such transformation requires an appropriate approach to be achievable. A systematic survey by [1] showed that many projects with smart solutions have been initiated to address existing thematic challenges faced by cities, such as parking issues, air pollution, traffic congestion, waste management, etc. It is worth noting that it is still unclear how many projects and at what integration level projects could take a city to a fully SC system level. In this study, we propose a view of an integrated model of a smart city system. The aim of this study was to illustrate a methodology through which an integrated city system can be modelled efficiently by keeping all the main components together to allow the operation of an integrated system within the boundaries of an SC. The objectives were to present and test a number of scenarios to illustrate how the proposed system would react under a set of particular conditions.

To achieve the aim and objectives of the paper, in Section 2, we describe and present previous studies to benchmark our work. In Section 3, we describe the proposed methodology and tool to model an integrated smart city system as well as the development process. In Section 4, we describe an integrated model of a smart city system and tests its behaviour, before offering conclusions and an overview on potential future work in Section 5.

## 2. Background: SC System Model—A Systems Approach

While some researchers are still trying to better understand how a city system can be redesigned [9], there is still no clear methodology that allows integration of all subsystems of an SC system. Many conceptual models have been proposed to accomplish such a task: [10] argued that proposed approaches in the literature were often not complete, not integrated, and non-communicating. They pointed out the absence of uniformity of the SC concept development, definition, and the lack of a methodology to evaluate developed models. They proposed a methodology for planning an SC through a model based on matrices; [8] focused on the interaction between a city and products services to propose a 'product-service system engineering approach' based on systematic engineering and sought to integrate the city service systems and product service systems; [4] built Information Technology (IT) to present a model made of a set of layers represented in a Geographic Information System two dimensional space; [11] looked at an SC system as composed of six layers which are green, interconnected, instrumented, integrated, intelligent, innovative, and presented a model which addresses global sustainability challenges in a local context; [12], motivated by the need to integrate a city system and human factor, presented a user-centred approach to design and model SCs building on application of systems engineering; [9] stressed the need for integration as a systematic answer to urban challenges, a need for tools to understand the complexity of the SC concept, and its capacity to solve urban challenges. In a case study on Vienna, the authors proposed an integrated conceptual model to respond systematically to urban challenges. In their study, they argued that an SC is an integrated and multi-dimensional system aiming to address challenges faced by cities and proposed a model that follows an approach linked to three main issues: the role of government and stakeholders, the role of displaying a comprehensive vision towards SCs, and SCs as a tool to tackle urban challenges; [13] worked towards the development of a platform infrastructure for a sustainable smart mobility subsystem. Their study proposed an information infrastructure architecture development method which presented a cycle from the development phase to the operation phase. The proposed platform focused on mobility information infrastructure and consisted of three components: the information collection component, the platform component, and the information provision service component.

The above previous study shows that although the necessity of integration of all six identified components [14] of an SC system is evident, it is still yet to be achieved. The presented previous work displayed the willingness to integrate SCs systems and make them work together as a unified whole and share information seamlessly. [15] observed contrasting views and the lack of an integrating view in an existing smart city system. It was observed that a smart city was a system that performs well in all smart subsystems which operate as interrelated entities and form a unified whole. [14] also showed that each subsystem of a city system had its own audience and services with a specific targeted subsystems to be transformed among the identified subsystems of a smart city system in [16]. Therefore, there was a disconnection of subsystems of a smart city systems and the need for system integration. Information and communication technology is at the forefront of a successful SC system [3] and there is a necessity of a well-integrated SC system—a set of elements/subsystems which interact with one another and which are viewed as a whole entity/system [17]. It is evident that there is a need to develop an efficient methodology which would allow the construction of a model of a fully integrated SC system that would allow all smart subsystems to share information.

## 3. Method and Tools

Given that not all complex systems are software systems, complex systems such as cities require methods and tools to be modelled, analysed and integrated. Model-Based Systems Engineering (MBSE) is a potential and robust methodology to model an SC system and is itself a contribution, and SysML, as tool, facilitates MBSE to add automation and enables analysis of a built system. MBSE could be a most viable option when it comes to SC system modelling and allowing information and technical communication across smart subsystems. MBSE can improve understanding of a system's needs and constraints, can ease the analysis while allowing to a bigger picture view when making big decisions and generating a whole system which is coherent and not dominated by a single perspective of a particular subsystem [18]. It extends beyond the engineering domains to support complex modelling and the integration of systems. In this study, MBSE was complemented by SysML, a tool capable of modelling a wide range of systems [19] developed by the Object Management Group [20]. It supports the practices of MBSE to develop solutions to complex and challenging problems such as analysis and multiple views visualisation of complex systems [17]. SysML itself is a standardised language and its advantages are that it is unambiguous, enforces consistency, prevents attempts at making incompatible connections, improves precision and efficient communication, is scalable, manages better complexity, and detect errors on any omissions in the early stage of systems development. [17] presented the taxonomy diagram which established the intended linkage between SysML taxonomy diagrams.

Like all other tools, SysML has limitations. It lacks the potential to independently execute mathematical expressions which are complex. In such a situation, co-simulation with other analysis tools such as MATLAB or any other compatible tool might be handy.

Model infrastructure is an important element to achieve the objective of system modelling work. Cameo Systems Modeller is a cross-platform collaborative MBSE by MagicDraw which provides a robust smart tool to define, visualise and track systems aspects and design processes [18]. This tool facilitates the visualisation of all aspects of systems in a standard way and complies with SysML diagrams. Diagrams included in SysML allow specification of system's structure and behaviour [21].

In this study, we developed an MBSE methodology to analyse a SC system and cope with its complexity through operational simulation. It focused on the conceptual design phase of a SC system and a model simulation was conducted to test it.

In general, system analysis can be done through simulation of models with or without a detailed system architecture. This study demonstrates that MBSE methodology and SysML can support SC system integration and analysis. To understand the system's operational side, emphasis was placed on system structure and logical architecture and focused on proper linkage between all subsystems of a city system and the system was executed under certain operational circumstances for analysis.

*System Model Development Process*

Figure 1 presents processes of the system model development. It displays the level of detail expected in the system, the system content and the outcomes which eases communication among all SC subsystems. It includes the six components (subsystems) [1] of a city system, an information platform (Figures 2 and 3), and users' benefits for easy information access from the information platform. The need for such integrated platform is driven by numerous factors, such as

- Better monitoring of the city,
- Gathering information from all sources for easy and fast access to information and fast reaction to incidents or any emergencies,
- Dynamically acting on urban population needs, and
- Having reliable information and services to the public for better decision making and efficient use of resources

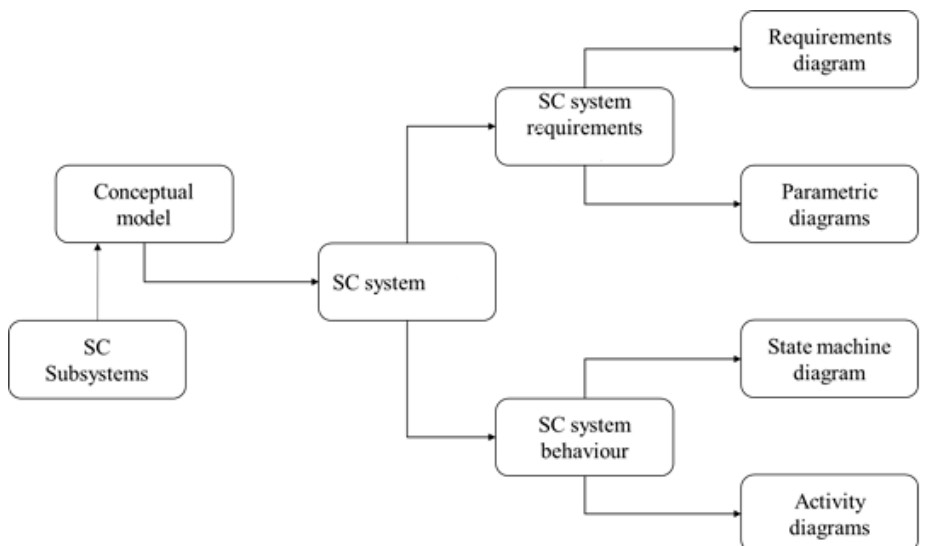

**Figure 1.** Model development process.

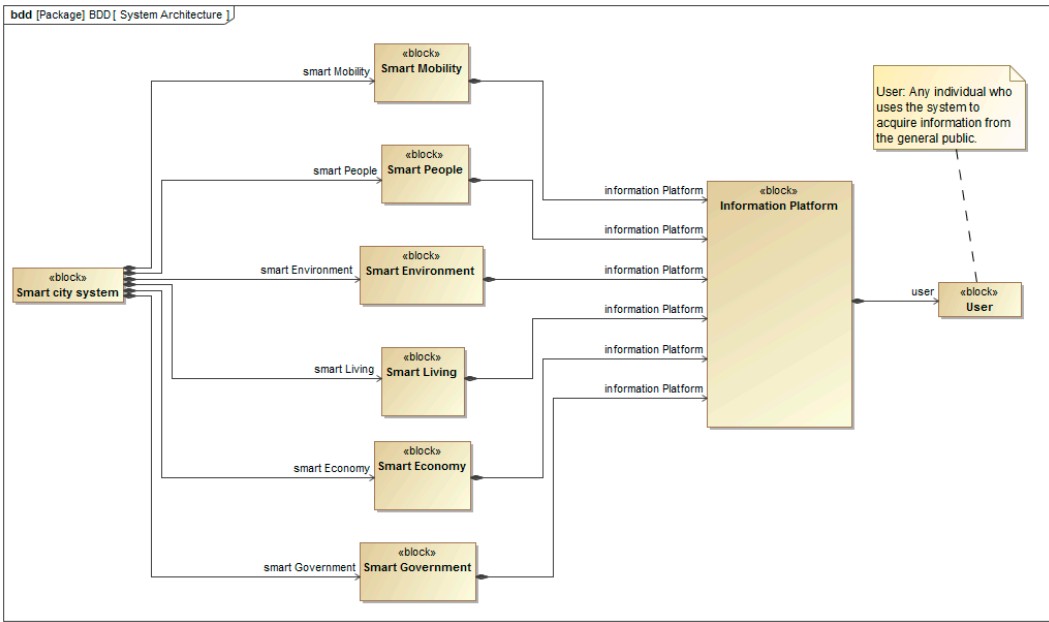

**Figure 2.** Smart city (SC) system model—logical architecture.

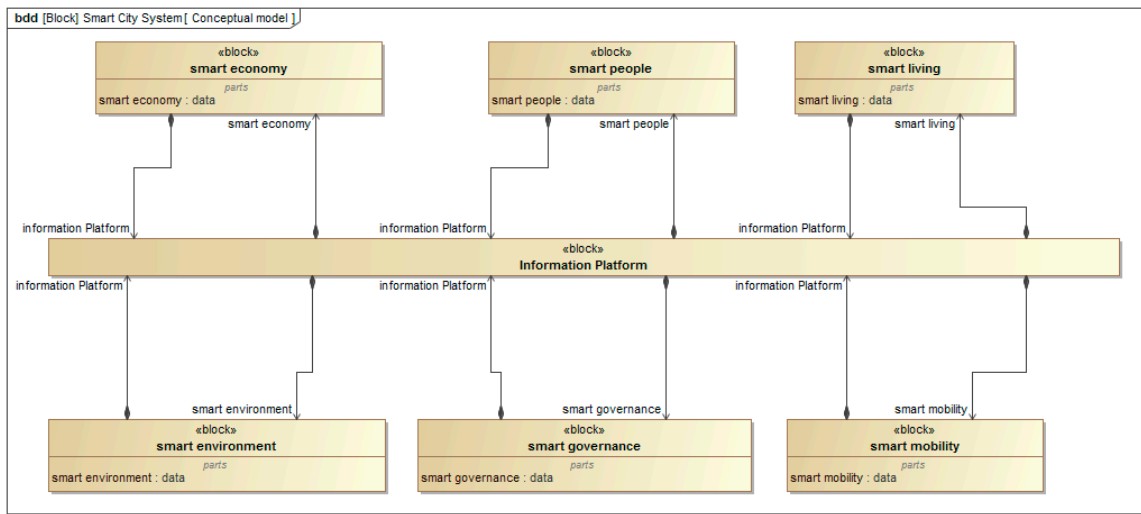

**Figure 3.** SC System model—conceptual structure—Adapted: [22].

Before information can reach the platform, data is collected from different sources and processed into meaningful information. However, this process was out of scope of this study and the focus was on information integration, the structure of the information platform and the behaviour of the model when subjected to certain scenarios.

## 4. Components and Logical Architecture of a Smart City System Model

In this section, an MBSE methodology to design and model a SC system with an integrated information platform is presented. Six components of a smart city systems were identified for the first time in [14] and were adopted by many researchers later [1]. As illustrated in Figure 2, the six identified component constitutes the subsystems of a smart city system. The structure of the conceptual model is illustrated in Figure 3.

The architecture of an SC system is defined in a SysML block definition diagram (BDD) (Figure 1). An important aspect of this model is the inclusion of an integrated information platform which is a gateway to easy information access within a smart city and allows interaction between all the smart subsystems of an SC system.

In the context of this study, the data was generated and processed into meaningful information at the level of each smart subsystem, as illustrated in Figure 3. The resulting information was then stored at the level of an information platform where it can be accessed. The information platform of an SC system is designed to focus on information sharing between the six smart subsystems of an SC as a method for an information integration process.

## 5. Concept of Operations and System Analysis

Often, in SysML modelling projects, there is a certain order which is followed for different diagram construction. The general order begins with the identification of system requirements which are illustrated in Figure 4 by a requirements diagram from which an activity or a state machine diagram was created to introduce the systems' behaviour and satisfy the requirements. Block definition diagrams were then created based on information in activity diagrams. From block definition diagram, parametric diagrams can then be generated. Figure 4 shows the requirements diagram of an SC system in which the main task of the system is described by the main requirement which is to develop a scalable SC information system which will integrate information from all levels of a SC system and make it easily accessible on the information platform.

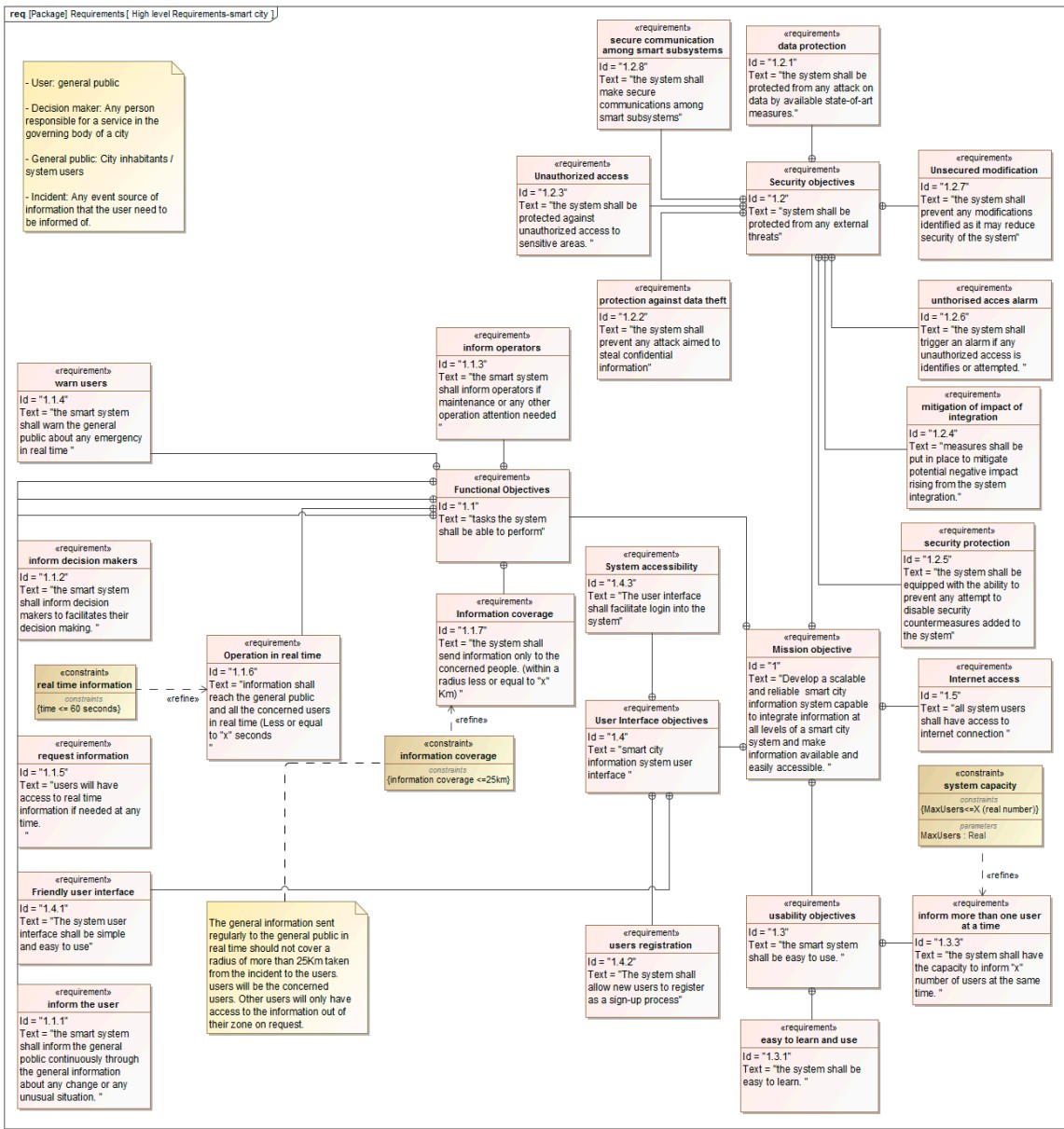

**Figure 4.** Initial high-level SC system requirements development.

Figure 4 presents the initial development of the requirements. It shows each requirement with an identifier ("id") and a descriptive text ("txt") to facilitate traceability. The diagram is decomposed into a set of sub-requirements which illustrate other relationships, such as refine, satisfy, and derive to support traceability. Identification of requirement is a fundamental part and critical factor to the success of system development [21]. There are various possibilities of classification of requirements, such as user requirement and system requirements. The requirement diagram belongs to SysML behavioural diagrams and its main purpose is to register all the relevant requirements of the system, to capture the hierarchies of requirement and relationships such as derivation, verification, satisfaction and refinement. The requirement diagram shown in Figure 4 illustrates the conditions to be satisfied and functions to be performed and achieved by a SC system. The diagram represents a hierarchy of requirements and depicts the requirements captured in a text specification and their inter-relationship. The diagram contains requirements for functional objectives, security objectives, interface objectives, and usability objectives.

Although the diagram in Figure 4 presents the high level SysML requirement diagram, not all aspects of an SC system operations are captured, but it does demonstrate common model operations and can be supplemented by an increase of details.

To understand how the system behaves, as presented in Figure 4, the state machine diagram illustrated in Figure 5 specifies the system states which are presented into a sequence of events which a system goes undergoes throughout its lifetime to respond to an event without deviating from the requirements needs.

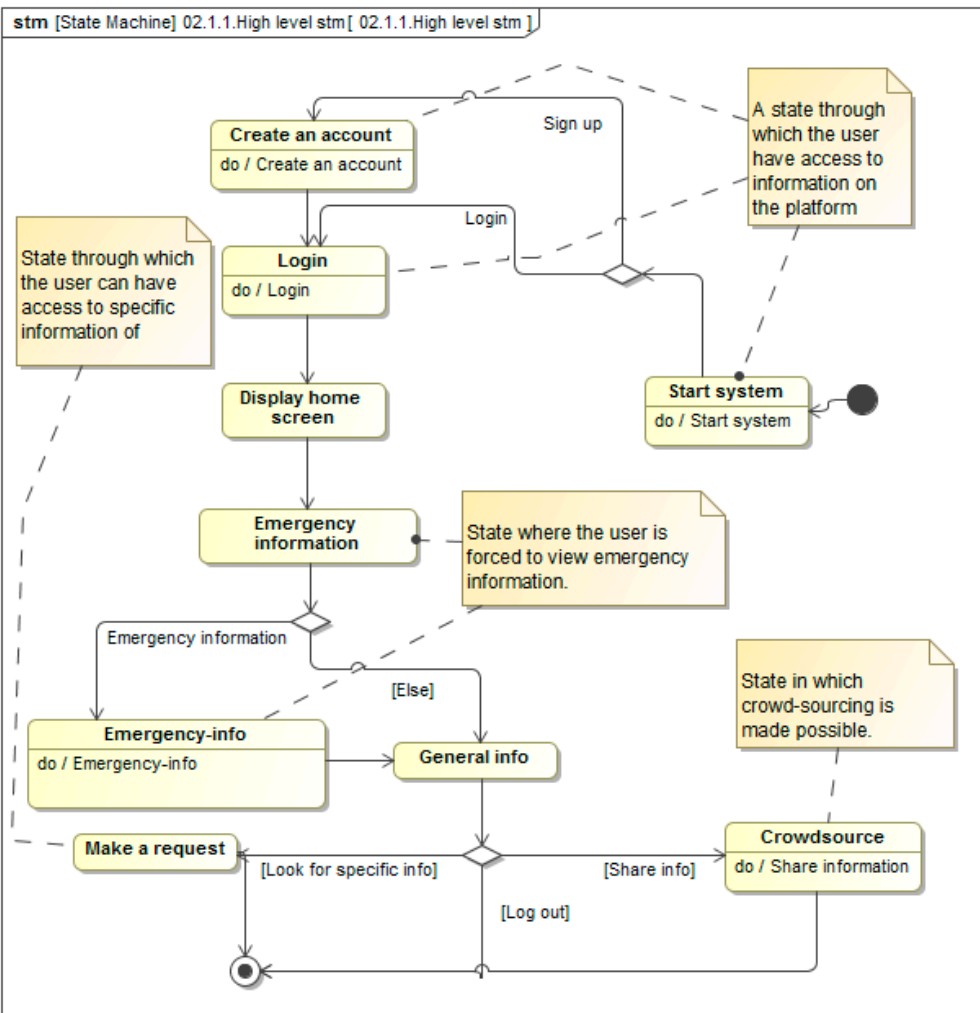

**Figure 5.** High-level behavioural state machine diagram for the information platform.

The triggers cause transition, which can be a signal, a change of condition, etc. A condition, "Guard", must be true for a trigger to initiate a transition and will act on a state and result in a change of state [17]. Figure 5 illustrates a state machine diagram which focuses on the information platform and defines how the system's states change as transitions are triggered. The diagram represents the behaviour of the system in terms of its transition between states which are triggered by events. For example, in Figure 5, the system cannot respond to event 'login' before 'sign up'. The figure illustrates the user's perspective and how the information platform behaves within the states it goes through once subjected to users' activities. It presents a high-level diagram where the top functions focus more on security, awareness of emergency situations, general information and crowdsourcing, which are normally in line with the requirement and the purpose of system development. Note that 'users' stands for the general public looking for quick response information.

The default state of the system is the 'start' state, illustrated by Figure 6a. This is the first action of the system in which the input is the user's details, which may require simple input of user identifiers if the user is pre-registered (Figure 6b), or the whole process of registration illustrated in Figure 6c, which would be the user's personal details, if he is a first time user.

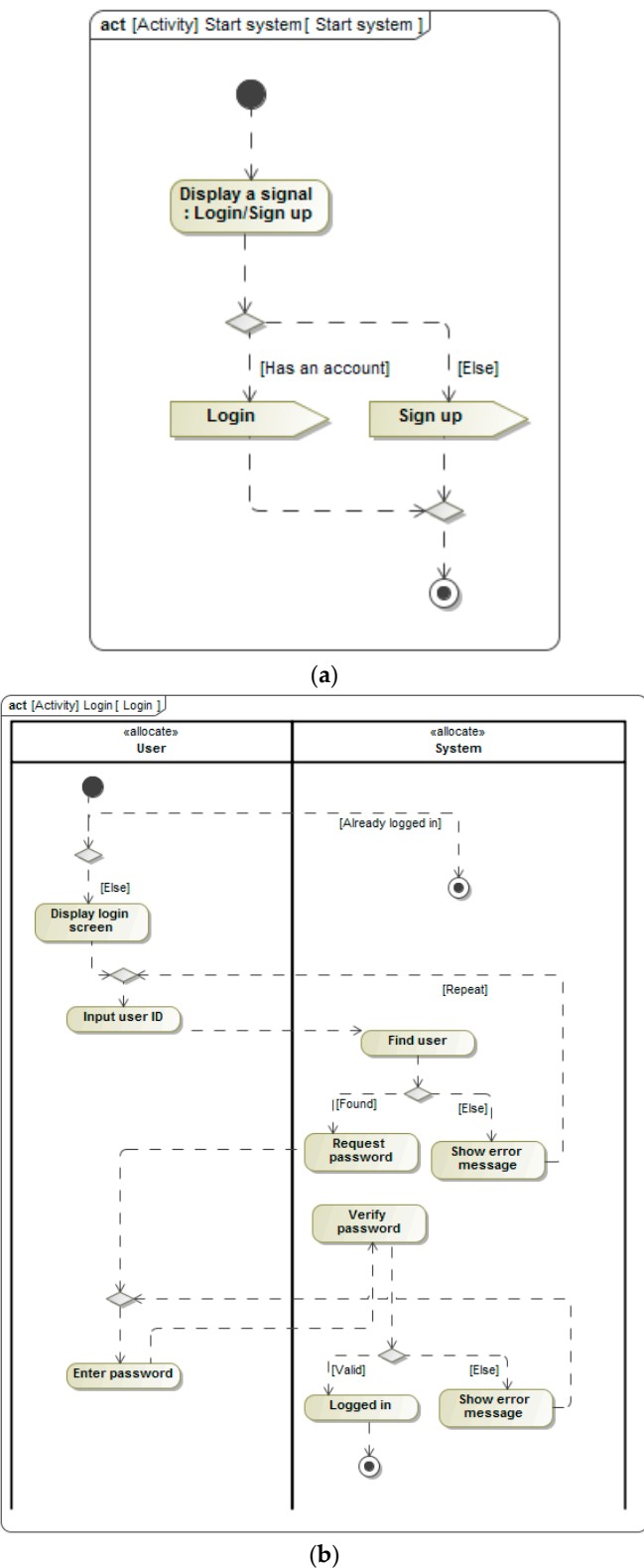

(a)

(b)

**Figure 6.** *Cont.*

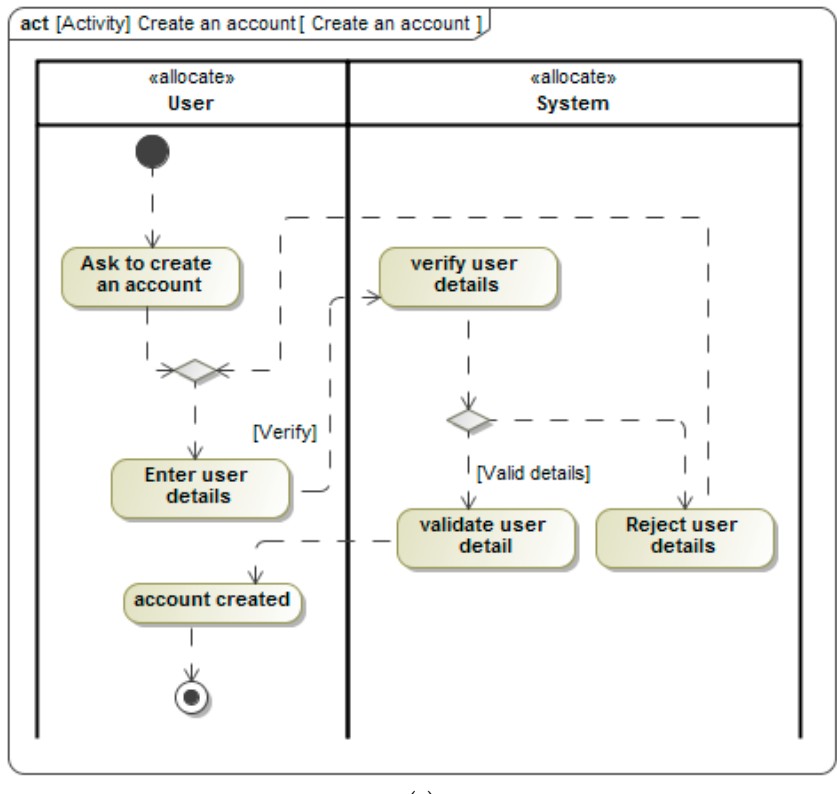

(c)

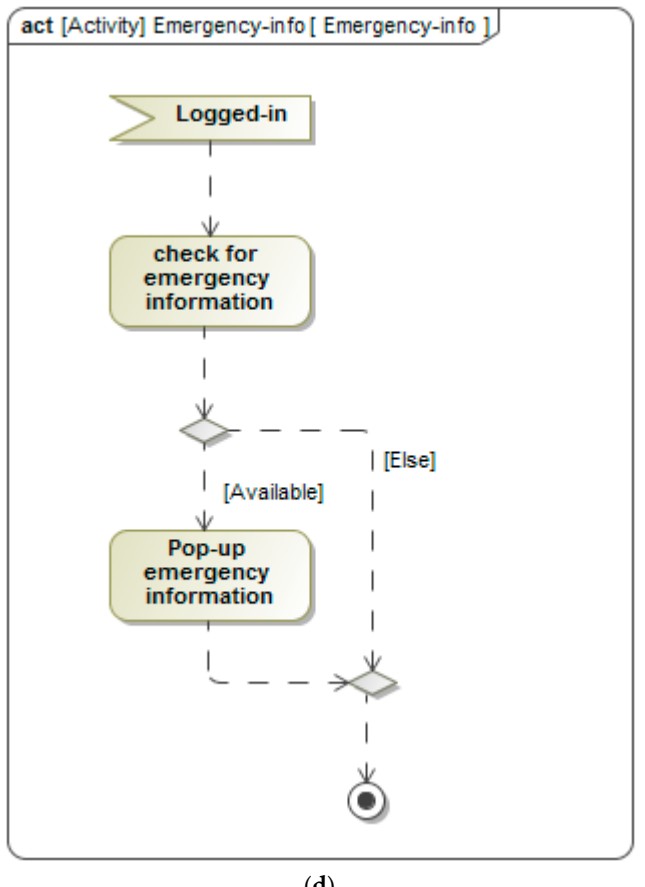

(d)

**Figure 6.** *Cont.*

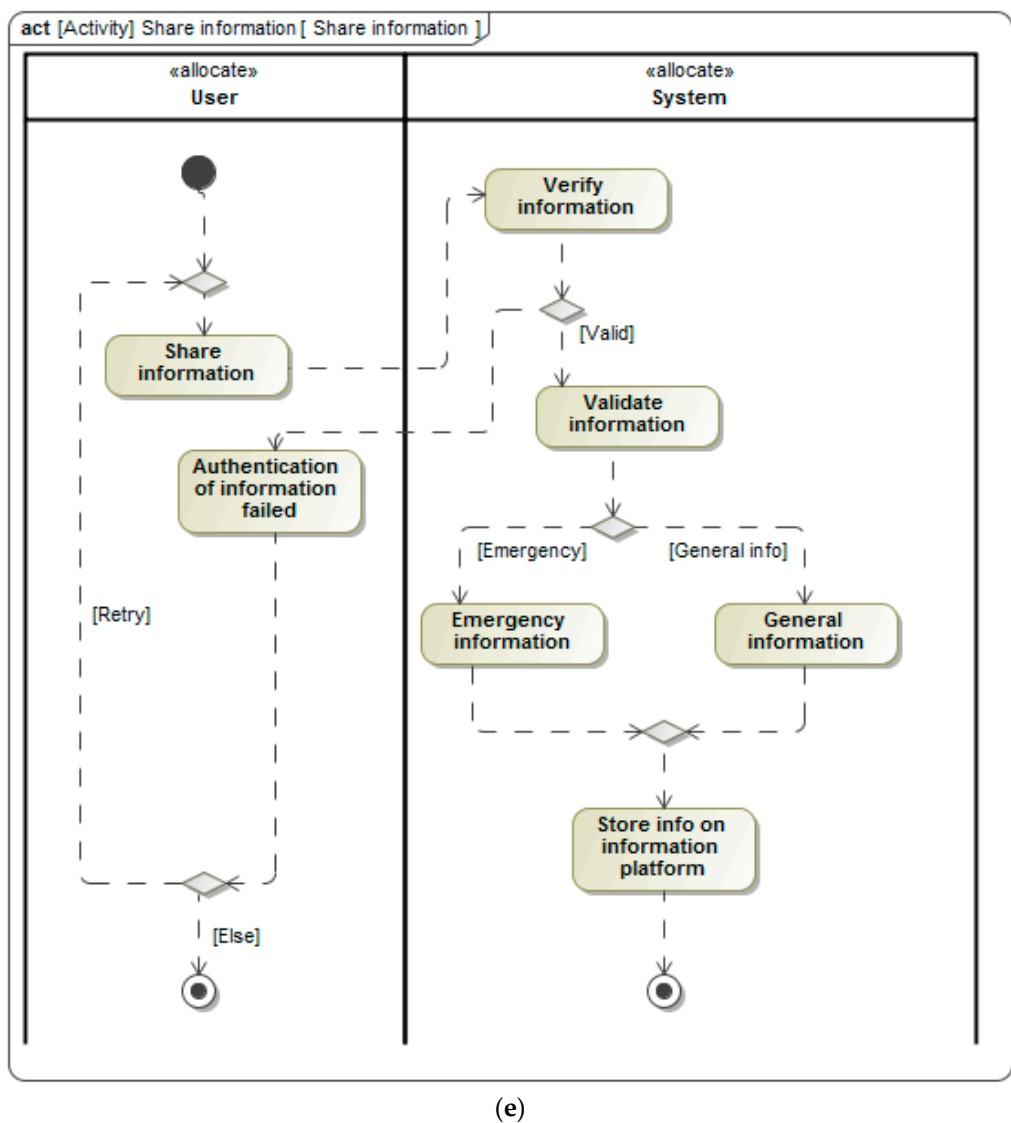

(e)

**Figure 6.** (**a**) Activities performed in 'start system' state of the SC system. (**b**) Activities performed in the 'Login' state of the SC system. (**c**) Activities performed in the 'create account' state of the SC system. (**d**) Activities performed in the 'Emergency-info' state of the SC system. (**e**) Activities performed in the 'share info' state of the SC system.

Once the user has been granted access to the information platform, as part of the transition to the next state, the system presents automatically relevant emergency information to the user shown in Figure 6d, which is information the user need to know, such as, in the case of mobility, traffic congestion building up due to an accident on the current user's route. The next state presents general information. At this level of the system, the user has three choices which can change the state of the system from general information to share information (Figure 6e) if the user wishes to share information as part of the crowdsourcing action put in place in the system, or from the general state, in which information judged relevant to user, depending on their location, is presented, or advanced search of information, if the user is looking for particular information not found in general information, or the logged-out state, if the user wishes to exit the platform.

The presented state machine diagram specifies the life cycle of the system in terms of states and transitions and used activity diagrams illustration in Figure 6 to illustrate the behaviour as specification of how the system responds to stimulus.

The presented activity diagrams are integrated in the composition system behaviour, which is captured in the state machine diagram presented in Figure 5. The activity diagram describes what the system must do to satisfy identified functions, shows the input actions of the users and the reaction of the system to the user inputs. When an activity is initiated, the execution starts at the initial node, and it then is initiated to the next activity until it reaches the final activity, and later, the final node. For the system (Figure 5), a set of actions are performed within each state as activities, which are executed (Figure 6) before the system can change to the next state. The presented activity diagrams include the semantics for precisely specifying the sequence of actions and a control flow represented by a dashed arrowhead line specifying the sequence of actions. Activity diagrams illustrate that the dynamic aspect of the system and are constructed in a manner that allows the execution of system through an engineering technique of forward and reverse. However, among their limitations, they do not have the capability to show the flow of messages among activities and do not give details of behaviour of collaboration among objects. Therefore, the activity diagram cannot replace the state machine diagram. The control flow provides constraints on when and in which order the action within an activity is executed and it does not start until the source action is completed.

The constraints are presented in blocks (Figure 7) as refinement actions of the system requirements diagram (Figure 4). Constraint blocks are special types of blocks with which equations can be defined and they have two main parts: parameters and expressions that constrain the parameters. This allows the support of parametric model construction [17], which can be represented using parametric diagrams. In general, system design requires engineering analyses to be performed. Such analyses may include reliability of performance of the system under certain scenarios. Parametric diagrams make it possible to capture constrains and create systems of equations to constrain properties of a block or particular requirements.

The functionalities of a system are mainly based on scenarios [21]. To analyse the behaviour of the presented model of an SC system, three scenarios are shown in parametric diagrams. The scenarios test the behaviour of the system when subject to capacity constraint, information delivery as time constraint and a constraint of defining the target of particular information. Figure 7a–c illustrates the constraints imposed on the system to test its functionalities. The presented constraints were adopted with the particular objective of analysing the system reaction when subjected to those three conditions. Figures 8–10 show parametric diagrams for the constraint blocks presented in Figure 7a–c. Given that constraint blocks do not show all the information that interconnects constraint properties, specifically, relationships between properties and parameters, this information is expressed by parametric diagrams through the connectors in Figures 8–10.

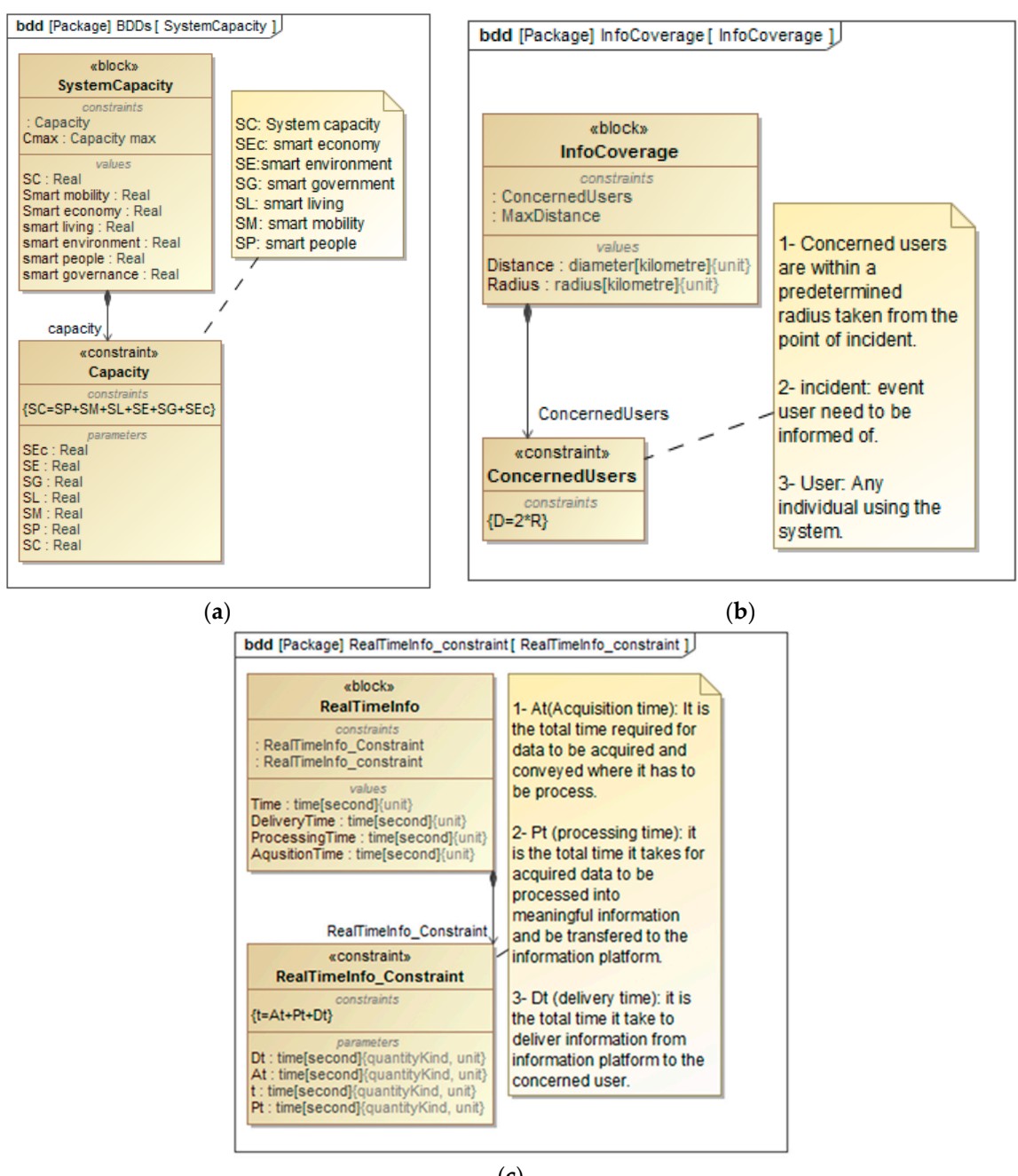

**Figure 7.** (**a**–**c**) Constraint block that defines parameters of equations for analysis.

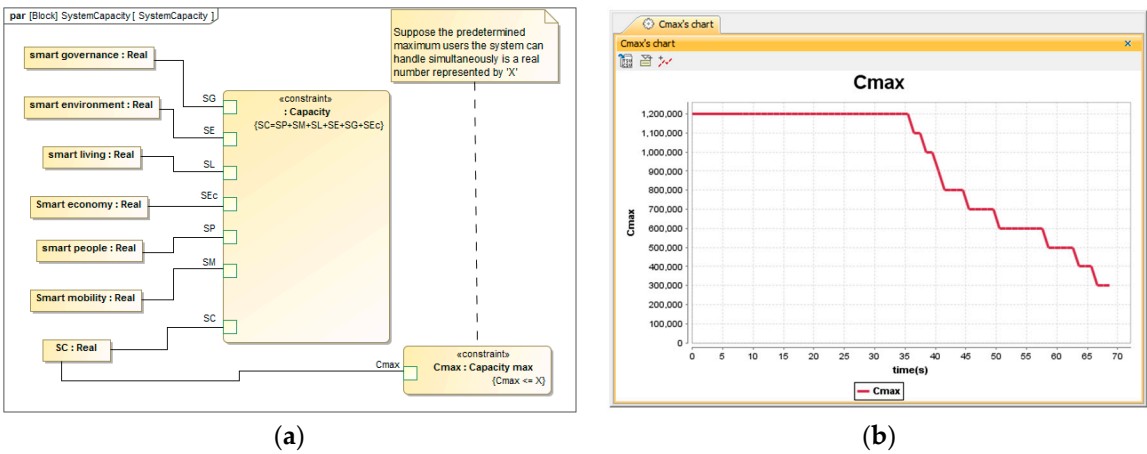

**Figure 8.** (**a**) Analysis of the capacity of the system (**b**) time series chart.

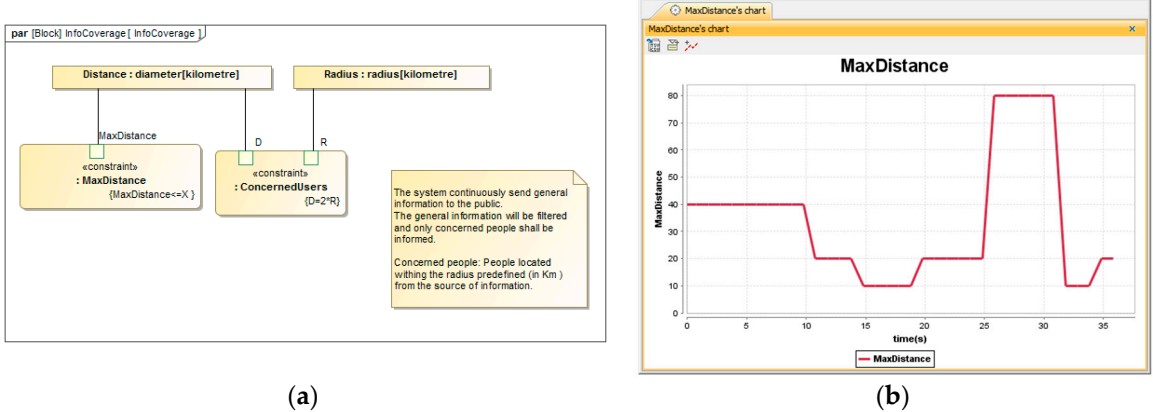

**Figure 9.** (**a**) Information coverage (**b**) Time series chart.

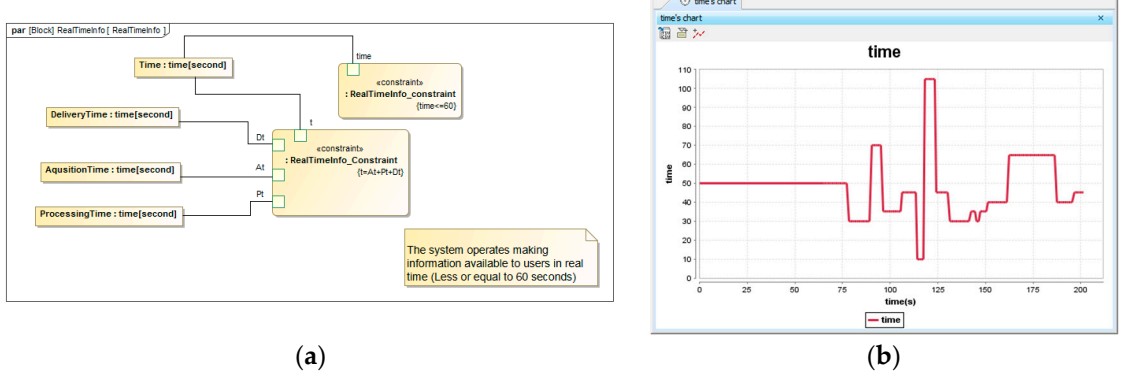

**Figure 10.** (**a**) Real-time information (**b**) Time series chart.

The first scenario presented in Figure 8 is concerned with the capacity of the system to focus on how many users the system can serve simultaneously (Figure 8a). The objective of imposing constrains on the system is to test whether the system is responsive to set requirements. The scenario considers that the system can have a limited number of users it is capable of serving simultaneously. The maximum capacity of the system is predefined and variables are inserted randomly to observe how the system reacts while operating under capacity, at capacity, and over capacity. The system continuously checks the capacity under which it is operating and signals a warning once the system reaches the maximum predefined capacity. A time series chart showing a continuous check of capacity

in time is displayed in Figure 8b. The results show that the system was responsive and satisfied the requirements.

Figure 9a shows that the information stored on the information platform comes from different sources and locations and only the concerned users, identified based on the distance between their location and the location of the incident, are to be informed of a particular incident if needed. This scenario tests the system by setting the concerned users at a particular distance from the source of information, which is, at the same time, the location of the concerned incident. Figure 9b is a time series chart which illustrates a continuous process of identification of concerned users to inform them.

The third scenario is illustrated in Figure 10a and tests the delivery of information in real time. The scenario set time to be considered as real time 't' (in seconds) which is the time required for the SC system to acquire data from an external source, to process it into meaningful information that can be understood by the users, and to transfer it to the users. Similarly to the previous scenarios, this process is continuous in time, as it is shown on the time series chart presented in Figure 10b. The scenario presented three parameters representing the time required to acquire data ($t_A$), the time required to process data into meaningful information ($t_P$), and the time required to deliver information to concerned users ($t_D$).

## 6. Conclusions

### 6.1. Summary

This study targets the information perspective of a smart city to highlight how an integrated model of a smart city system can be developed using MBSE methodology. A review of prior research revealed that the purpose of the process of transformation of traditional cities into SC was to deal with the present challenges faced by the city and to counter future challenges. Multitudes of approaches and initiative aiming to model and transform traditional cities into SCs have shown contrasting views and lacked an integrative view. Modelling complex systems such as the SC system can be challenging, but with the MBSE method powered by the SysML, a graphical modelling language can help with such a task.

Recalling that this study was motivated by the necessity to produce a methodology to model an integrated SC system, the results presented herein initiated procedures that began with an initial high-level requirement diagram of a smart city system and those requirements were used to initiate the external design of an experimental model simulation for system analysis. The presented model of a smart city system strengthens linkage in the architecture of an SC system development and system analysis. Procedures that use SyML products to support MBSE methodology and development of external structure of a smart city system model and simulation for analysis were described. This study demonstrated how this methodology could bring solutions to complexity with regards to SC system modelling and integration of information for easy access and effective operation. Using MBSE, SysML, and a Cameo Systems Modeller, the presented methodology sets the stage for how to model a system of an SC and address the necessity to have all its subsystems interrelated, connected and integrated. The analysis performed on the model through scenarios was for testing the purpose to understand the behaviour of the proposed model, interaction of elements, and verification of how the system responds when subjected to particular events. The presented integrated model allowed us to gather information from all the smart subsystems of an SC system on an information platform which supports multiple use cases. Given that the defined and described methodology links the SC system architecture and analysis, the proposed system model can be a starting point for cities to develop their own integrated model and platform in their specific context.

### 6.2. Potential Future Research

There are numerous potential research strands related to this work that would further extend SC system model understanding and integration. More directly, the next contribution would detail

more the described MBSE methodology for SC system integration focusing on the internal structure and architecture. More potential future studies would take further the development, analysis, and verification of the presented methodology to model SC systems through simulation of more practical and realistic scenarios focusing on interaction and integration of different subsystems to develop an easy interpretation to stakeholders, to highlight the value of eased communication and to deal with the contradicting views observed in prior research. MBSE-SysML integration with other tools, such as MatLab, to perform co-simulation would also be considered for further analysis and validation of the mode. This would certainly initiate new challengers about behaviours, management and simulation of a unified and integrated model of an SC system.

**Author Contributions:** Conceptualization, J.M. and T.B.; methodology, J.M. and T.B.; software, J.M.; validation, J.M. and T.B.; formal analysis, J.M. and T.B.; resources, J.M. and T.B.; writing—original draft preparation, J.M.; writing—review and editing, J.M., T.B., K.J.B. and S.B.M.; visualization, J.M.; supervision, T.B., K.J.B. and S.B.M.; project administration, J.M. and T.B.

**Funding:** The research was funded by GCU/Rwanda.

**Acknowledgments:** Many thanks to Mario Cools for making LEMA's infrastructure accessible to facilitate this research.

**Conflicts of Interest:** The authors declare no conflict of interest.

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
