# Peer review of "A Methodology to Model Integrated Smart City System from the Information Perspective"

_smartcities, doi:10.3390/smartcities2040030_

Round 1

Reviewer 1 Report

The paper: Conceptual model of a Smart City: A methodology to model an integrated SC system from the information perspective has been very well structured, researched, discussed and concluded. However, a few inputs would make it better that its current state.

1. Lines 145-150, elements of SCs, more citations could make it better: I have recommend some herein below.

Tomaszewska, E. J., & Florea, A. (2018). Urban smart mobility in the scientific literature—bibliometric analysis. Engineering Management in Production and Services, 10(2), 41-56.
Balasubramaniam, A., Paul, A., Hong, W. H., Seo, H., & Kim, J. (2017). Comparative Analysis of Intelligent Transportation Systems for Sustainable Environment in Smart Cities. Sustainability, 9(7), 1120.
Freudendal-Pedersen, M.; Kesselring, S.; Servou, E. What is Smart for the Future City? Mobilities and Automation. Sustainability 2019, 11, 221.
Bamwesigye, D.; Hlavackova, P. Analysis of Sustainable Transport for Smart Cities. Sustainability 2019, 11, 2140.
Al-Thani, S.K.; Skelhorn, C.P.; Amato, A.; Koc, M.; Al-Ghamdi, S.G. Smart Technology Impact on Neighborhood Form for a Sustainable Doha. Sustainability 2018, 10, 4764.

I highly recommend the paper for publication with minor revision.

Author Response

P.S. English spell was re-checked. 

Reviewer 2 Report

I have a rather ambivalent view of this submission, for most of its individual components are satisfactory, yet as a whole it does not seem to deliver what it promises. The title refers to a conceptual model of a smart city, but the paper provides a rather sketchy discussion about such a model and its components. Also the methodological discussion remains rather formal and superficial, even though it must be assessed within the requirements of systems engineering methodology, which communicates with engineers rather than with social scientists. In brief, the paper should be improved by focusing on research problem, structure and language. It has a potential, but it is in its current from incoherent. Note: the application and relevance of MBSE and SysML as they are utilized here in creating/exploiting a domain model must be assessed separately by experts of systems engineering methodology or those who are familiar with model-based approaches in engineering. Below are my detailed comments.

First, regarding the title, it may be worth considering whether this paper really produced a conceptual model of a smart city. In its current form much effort is devoted to methodology rather than the model itself.

Abstract would benefit from rewording some expressions. Is smart city really that "new" concept? And do not write "Smart City (SC)"; abbreviation should be presented simply as ... smart city (SC). 

Introduction must be rewritten. The problem is that it does not really help a reader to get your point. It repeats key concepts in a vague way and the real issues stay behind dry jargon. Try to avoid vagueness, passive expressions and truisms, and focus on your key points, and explicate them and their relevance to a reader. Technology, flow of information, "appropriate approach" etc. are important, indeed, but in this kind of article you should provide something smarter than just a few truisms. A matter that needs to be addressed and illustrated, for example, is what the integration actually means in the given context and what kind of SC components are integrated. You should try to make sense of the issue addressed in your paper. In addition, you must formulate a clear research problem at the end of the introduction, and after that, describe the structure of the article.

Sections 2 and 3 form a theoretical framework for your empirical analysis. Section 2 would benefit from accuracy and rigor regarding the description of SC layers and components. This may be a matter of taste, but to me the ideas like SC platforms, Cyber-physical Systems and urban service ecosystems are essential part of this picture. And when you describe application areas, try to be as precise as possible (instead of "governance, environment, mobility, etc."). Later in the analysis you seem to use the six categories of Smart City Wheel, which is not necessarily that systematically built model. Whatever is the case, you should be systematic in determining the categories that you see essential in describing and theorizing SC as a system.

Regarding section 3, I am not sure that the demarcation you make between 'smart' and 'intelligent' is justified (e.g. Komninos uses these terms interchangeably). And the rankings per se should not be the issue here at all. Instead, we need categories that are rationally or systematically developed. Now the discussion about domains is rather random, which is not a good basis for a model-building endeavour.

Methodological section could be more operational or methodical in terms of the use of MBSE and SysML. The motivation of the system model development (lines 199-206) should have actually be done much earlier in the theoretical framework. Section 5 should concentrate of building the model. The six components discussed in Figure 3 were not sufficiently defined and justified in the theoretical framework. I am not sure of the relevance of discussion from Figure 5 onwards (from Figure 5 to Figure 6e), until you incorporate the SC components into your discussion. In general, discussion in section 5.2 includes a lot of material that I consider irrelevant. This may be, however, a matter of taste (or that of a discipline), why it is fair to leave a reservation to this judgement.

Concluding section does not explicate the challenges and solutions to the building of smart city model sufficiently. It is even difficult to grasp what the “proposed model” actually is, as discussion was more devoted to describing the process and operations. In brief, try to be more specific when describing the results of your conceptualization and modelling, and assess how they relate to prior research. Pay attention to the “added value” or new knowledge that your contribution has generated.

Minor points: polish up your text; 'smart city' should be written with small letters like any noun. "Smart City (SC)"? (line 35); SCs concept? (47); repetition of "one that" is disturbing (49, 51, 54); some figures are hard to read due to the small size of letters; capitalized letters in authors’ names in reference [9] (line 425).

Author Response

P.S. English changes were made. 

Reviewer 3 Report

Lines 147 to 167 need a review and six dimensions identified.

Sections 4 and 5 need improvements to establish a link to the topic and present the modelling method in a coherent manner in order not to lose readers interest. 

There are no clear outcome from modelling exercise eg case study. 

Author Response

Note: English spell check was made. 

Round 2

Reviewer 2 Report

Revision looks sufficient. The paper is in its current form more coherent than the first submission. The revised version is publishable. Before submitting final version, it is always good to take a final check in order to make sure there are not typos or other mistakes in the paper.

Reviewer 3 Report

Further review by other experts might be required to assess the paper's content and significance.